# Swiss Medical Devices for Autologous Regenerative Medicine: From Innovation to Clinical Validation

**DOI:** 10.3390/pharmaceutics14081617

**Published:** 2022-08-02

**Authors:** Farid Gomri, Solange Vischer, Antoine Turzi, Sarah Berndt

**Affiliations:** 1Regen Lab SA, 1052 Le Mont-sur-Lausanne, Switzerland; fgomri@regenlab.com (F.G.); svischer@regenlab.com (S.V.); aturzi@regenlab.com (A.T.); 2Department of Plastic, Reconstructive and Aesthetic Surgery, Faculty of Medicine, Geneva University Hospitals, 1205 Geneva, Switzerland

**Keywords:** platelet-rich plasma, PRP, tissue regeneration, cell therapy, regenerative medicine, autologous biologics, ortho-biologics, hyaluronic acid, HA, good manufacturing practices, GMP, medical devices

## Abstract

Regenerative medicine, based on the use of autologous tissues and embryonic, stem or differentiated cells, is gaining growing interest. However, their preparation, in a manner compliant with good practices and health regulations, is a technical challenge. The aim of this manuscript is to present the design of reliable CE marked medical devices for the preparation of standardized platelet-rich plasma (PRP) and other autologous biologics intended for therapeutic uses. There are numerous PRP isolation processes. Depending on the methodology used, PRP composition varies greatly in terms of platelet concentration, platelet quality, and level of contamination with red and white blood cells. This variability in PRP composition might affect the clinical outcomes. The devices presented here are based on a specific technology, patented all over the world, that allows the precise separation of blood components as a function of their density using thixotropic separator gels in closed systems. This allows the preparation, in an automated manner, of leukocyte poor PRP with a standardized composition. Production of different forms of PRP is a clinical asset to suit various therapeutic needs. Therefore, we are offering solutions to prepare PRP either in liquid or gel form, and PRP combined with hyaluronic acid. These biologics have been successfully used in many different therapeutic domains, resulting in more than 150 published clinical studies. We also developed the CuteCell technology platform for cell culture expansion for further autologous cell therapies. This technology enables the safe and rapid in vitro expansion of cells intended for therapeutic use in good manufacturing practices (GMP) and autologous conditions, using blood-derived products as culture media supplementation. We summarize in this article our 20 years’ experience of research and development for the design of PRP devices and, more recently, for PRP combined with hyaluronic acid.

## 1. Introduction

Major developments in regenerative medicine based on autologous tissues and cells (embryonic, stem or differentiated) have occurred since the beginning of the 20th century. Reconstructive and orthopedic surgery techniques use the patient’s own tissues, such as skin, fat, tendon or bone, to treat trauma or inherited defects. More recently, the idea of using cells or growth factors to stimulate tissue regeneration at the site of the lesion has emerged.

Platelet-rich plasma (PRP) therapy has gained popularity since the earliest reports of its clinical use in the 1980s and 1990s [1]. David R. Knighton described the first use of “locally acting growth factors obtained from human platelets and applied topically”, highlighting how interesting it may be to isolate growth factor-secreting plasma cells from whole blood to induce tissue regeneration, particularly in chronic wound healing. At that time, Knighton used laboratory techniques to prepare PRP [2,3].

PRP is easily obtained, as it requires only a venipuncture. Blood components are then separated by centrifugation to obtain the fraction containing the plasma and the platelets. The relatively low cost and ease of use have facilitated PRP’s rapid expansion within medical practices [4,5]. PRP preparation has been greatly simplified in recent years, thanks to the development of commercial PRP preparation devices [6]. These devices also allow the preparation of PRP in conformity with health regulation and good practices requirements.

As there are many publications in which PRP was prepared using laboratory accessories, or tubes intended only for diagnostic use, most health professionals are not aware that PRP intended for therapeutic use should be prepared only with certified medical devices intended for PRP preparation, see, for example, section 201 (h) of the Federal Food, Drug, and Cosmetic Act, or article 5 of the 2017/745 regulation on medical devices, for the USA or European Union, respectively [7,8].

## 2. Standard PRP Isolation Processes

There are numerous PRP preparation protocols, differing by preparation devices, centrifugation conditions and operator dexterity, so, as a consequence of this, PRP is used to qualify biological products that vary greatly in their platelet concentration, quality and content in growth factors, and level of contamination with red blood cells and pro-inflammatory white blood cells [9]. This large variability in PRP preparations, and the different treatment protocols, create a challenge when trying to draw conclusions from the literature on PRP clinical benefit. This, in turn, has prompted the development of many PRP classification systems to facilitate reporting of clinical investigations [1,10].

However, the different technologies available produce mainly two types of PRP: plasma PRP and buffy coat PRP [10] (see Table 1).

Plasma PRP is generally produced using low centrifugal force and/or very short centrifugation of a small volume of blood (usually less than 20 mL). In these centrifugation conditions, platelets are still floating in the plasma at the end of the centrifugation. They form a gradient, with the highest concentration of platelets close to the layer of white blood cells that is called the buffy coat. With this technique the PRP is usually a leukocyte poor PRP (LP-PRP) with virtually no red blood cell contamination and a platelet concentration factor less than 3 times (3×) the baseline value in blood [10]. For some devices, a second centrifugation is performed to increase the final platelet concentration over 3×, by discarding some platelet poor plasma (PPP). Hence, these devices are required to process a higher blood volume (60 mL) to obtain 6–7 mL of highly concentrated LP-PRP [11].

However, the composition of these LP-PRPs is highly operator dependent. As there is no physical barrier between the plasma and the buffy boat, the PRP can easily be contaminated with blood cells if the operator gets too close to the buffy coat. On the other hand, if the operator stops collecting the plasma too far from the buffy coat, then the platelets with the highest density will not be recovered. The densest platelets are the ones with the richest content in growth factors [12]. Thus, such LP-PRP may contain less growth factor than other PRP with the same platelet concentration, prepared using a technique that allows more efficient recovery of platelets with highest densities.

Buffy coat PRP is produced using high force centrifugation which concentrates the platelets at the level of the buffy coat with the white blood cells. Depending on the devices, either the full buffy coat, or a part of it, is collected in variable volumes of plasma. The resulting PRP is leukocyte rich PRP (LR-PRP) with a platelet concentration factor over 4×. It contains concentrated pro-inflammatory white blood cells and a variable level of red blood cells and, thus, is often reddish [10]. A high volume of blood (25–60 mL or more) is required to produce 1 to 6 mL of LR-PRP, as PPP needs to be discarded to reach a platelet concentration factor over 3×.

Buffy coat devices with an hourglass shape are designed to collect the full buffy coat layer or most of it. The operator, using the screwing system of the device, should move the buffy coat zone in the narrowest part of the device, before collecting it. The resulting PRP composition is, therefore, operator dependent and contaminated with highly concentrated pro-inflammatory white blood cells [13].

Devices that contain a floating shelf or buoy with a specific density, allow physical separation of the blood components during centrifugation. The platelets and the plasma, that have a lower density than the physical separator, are recovered in a separate compartment of the device, from which the PRP is collected. Although the physical separator allows the retention of most red and white blood cells, the resulting PRP is, nevertheless, a LR-PRP with a variable level of red blood cell contamination and, thus, is a pro-inflammatory PRP [14].

Automated computer-assisted systems are smaller versions of apheresis machines used in blood donation centers for the preparation of platelet concentrates intended for transfusion. After separation by centrifugation with variable g force in a specific machine, blood components pass through light sensors. Change of color and turbidity in the sample triggers a switch that sends blood components to different bags or to a syringe. With this kind of device, the resulting PRP composition is setting dependent. The operator programs the device according to the desired type of final product. However, it is not possible to set the level of cellular contamination below a certain threshold. For example, the minimal setting for the hematocrit is 2% in one type of device [14]. Consequently, PRP produced with these devices are usually leukocyte rich PRP. It is to be noted, that using settings that minimize red and white blood cell contamination, induces a significant decrease of the platelet recovery (<50%) and probably also the loss of the densest platelets [14].

## 3. Manufacturing Innovative Devices for Standardized PRP Preparation

To fulfill the need of standardized PRP preparations, Regen Lab, a Swiss pharmaceutical and medical device company, has developed complex polymer-gel separation systems, that allow the efficient recovery of platelets and plasma and deplete red and white blood cells in an automated closed-circuit system. This innovative technology combines the advantages, without the disadvantages, of both buffy coat and plasma PRP preparation methods. These devices respond to the different challenges of efficient PRP preparation in accordance with international medical device regulations, which means that the devices are effective for PRP isolation and safe for patients and operators. There is also the implication that the manufacturer has to be compliant with all norms and requirements linked with the manufacturing of medical devices. In addition, these devices should respond to the clinicians’ needs in various therapeutic domains.

### 3.1. Design and Manufacturing of Medical Devices for Safe and Effective PRP Preparation

#### 3.1.1. Essential Requirements for the Manufacturer

In addition to international medical devices regulations, a manufacturer of medical devices should follow many norms and much guidance. Regarding ISO norms, the list is not intended to be exhaustive. The manufacturer should have a quality management system that is certified according to ISO 13485 and a risk management system (ISO 1471). It should perform continuous clinical evaluation (ISO 14155) and post market surveillance (ISO/TR 20416) to verify the safety and performance, including clinical benefits, of the device when used as intended by the manufacturer. The manufacturing processes should be validated. The manufacturer should ensure that its devices are manufactured in a controlled environment, e.g., in clean rooms (ISO 14644, ISO 14698) with approved materials, either pharmaceutical grade or tested for their biocompatibility (ISO 10993). In addition, the devices for PRP preparation should be manufactured to be sterile (ISO 11737, 17665, 11137), packaged in such a way that the sterility is maintained all along the shelf life of the devices (ISO 11607) and adequately labelled (ISO 15223).

Then, in order to be placed on the market, a medical device should be approved by health authorities in each country where it is marketed. Regulations differ in each country but the main requirements for approval are always safety and effectiveness of the devices. In the European Union, there is a unique regulation (Medical Device Regulation MDR 2017/745, that is replacing the Medical Device Directive MDD 93/42 EEC) applicable for all members and for other countries outside the Union, such as Switzerland, that have decided to follow this regulation. Under this regulation, medical devices for PRP preparation are classified IIa or IIb and should be certified through a notified body. Health authorities should also perform a regulatory audit of manufacturers. To simplify this process, the Medical Device Single Audit Program (MDSAP) has been implemented. It allows a single regulatory audit of the device manufacturer, by a recognized auditing organization that satisfies the relevant requirements of the regulatory authorities participating in the program. The members of the MDSAP program are Australia, Brazil, Canada, Japan and the USA. Other countries, such as the United Kingdom and the European Union, are, for the moment, only observers. Our quality management system is certified under this program, as MDSAP certification has been mandatory for marketing medical devices in Canada since 2019.

#### 3.1.2. Design of Devices That Respond to User Needs for Preparation of Standardized PRP

The manufacturer should determine the intended use and the functional requirement specifications for its devices. In this case the devices are intended for the preparation of standardized autologous PRP, and the main functional requirements are that the devices be sterile, single use devices designed to be used at the bedside of the patient by physicians to obtain a small amount of venous blood. 


**Sterility:**


The medical devices are packaged in individual blisters and are sterilized by exposure to a minimum dose of 25 kGy gamma irradiation, or, for the devices that contain hyaluronic acid, by moist heat.

To maintain the sterility of the biological sample, a device intended for PRP preparation at the bedside of the patient should work in a closed circuit (Figure 1). This also has the advantage of minimizing the risk of blood exposure for the operator. We have opted for evacuated tubes. This has the advantage of allowing the collection of a pre-set volume of blood directly into the device, using a standard blood collection set with a tube holder. 

Once the PRP is prepared it is collected with a syringe connected to a transfer device. The use of these accessories ensure that the biological sample is not exposed to air and reduces the risk of microbial contamination. The tube holder and the transfer device are equipped with an internal needle that does not core the stopper of the tube. Thus, when the tube is removed from the accessories it is still airtight. These tubes are manufactured with pharmaceutical grade (Ph. Eur &USP) type I borosilicate glass with cerium and closed with bromobutyl rubber stoppers, tested for their biocompatibility according to ISO 10993 norms.


**Anticoagulant:**


To keep the PRP in liquid form until used, a PRP device should contain a reversible anticoagulant. It is possible to prepare PRP without anticoagulant; however, the device should be in plastic and the PRP remains fluid only for a short time. The use of a reversible anticoagulant is more convenient, provided that the anticoagulant has no ancillary effect on the patient. Sodium citrate was chosen as citrate-based anticoagulants are fully reversible. In comparison to ACD-A (acid citrate dextrose solution A) which is an acidic (pH 4.5–5) anticoagulant often used for PRP preparation, sodium citrate has the advantage of having a neutral pH (pH 7) and contains no sugar [15]. Sodium citrate is more physiological and has no ancillary effect on the patient. It is pre-dosed in the tubes, thus avoiding the risks involved with manipulation of the anticoagulant by the operator.


**Standardized PRP preparation with separator gels:**


The main challenge when designing a device for PRP preparation is to find an effective technology to isolate the PRP. In order to be operator independent, a physical separation is needed to mechanically isolate the platelets and the plasma from the other blood components. As mentioned above, other devices on the market have been designed with floating shelves or buoys, or use computer assisted systems with light sensor and valves, to physically separate blood components. In comparison to these technologies, the use of polymer separator gels has many advantages. It can be used in small blood collection tubes, thus reducing the volume of blood needed, the gel separation requires only a short centrifugation (5 or 9 min, depending on the type of gel), and it allows the precise separation of blood components at the cellular level [16].

These separator gels are designed with a specific density; they are lighter than the unwanted red and white blood cells and heavier than the platelets and the plasma. These gels also have thixotropic properties that allow them to become fluid when submitted to a centrifugal force of 1500× *g*, and to regain their original consistency when the centrifugation stops [16]. Separator gels used in our devices are biocompatible, according to ISO 10993 norms, and chemically inert and, thus, safe for the patient.

During centrifugation, the blood components are separated according to their specific densities and form distinct layers that are the plasma, buffy coat and red blood cell pellet. The buffy coat is a whitish zone that contains the platelets and the white blood cells. The separator gel, thanks to its thixotropy, becomes fluid, migrates upwards in the device and intercalates itself precisely inside the buffy coat layer at the level of its own specific density. At the end of the centrifugation, the separator gel regains its solid consistency and forms a solid barrier that mechanically separates the blood components. It isolates the platelets and the plasma in the upper part of the tube, while the undesired red and white blood cells are trapped below the separator gel in the lower part. Due to the centrifugal force, the platelets form thin sediment on the upper surface of the gel. To obtain PRP the tube should be gently agitated to put the platelets back in suspension in the plasma (Figure 2). At this point, the preparation is ready to be used by the physician.

As it is the polymer gel that separates blood components according to their density, PRP isolation in such devices is specific and operator independent, and is, thus, reliable and reproducible. The resulting PRP is a standardized plasma PRP with a high platelet recovery (>80%) with no specific loss of the densest platelets, a low level of leukocytes, with a specific depletion of the pro-inflammatory white blood cells and virtually no red blood cells. Red blood cells are undesired in PRP as their degradation releases free radicals that induce oxidative stress and components, such as heme from hemoglobin, that are deleterious for cells [17,18].

Leukocytes can be divided into three main populations in blood: granulocytes (65%), lymphocytes (30%) and monocytes (5%). Different types of separator gels that differ in their specific density have been developed. This allows either the recovery of only the platelets (RegenBCT/A-CP) or the platelets and the mononuclear white blood cells (lymphocytes and monocytes) (RegenTHT) with a high efficacy. Even in the latter case, the resulting PRP is still a LP-PRP as the final total white blood cell concentration remains below the baseline level in blood. Depending on the type of separating gel, the recovery of mononuclear white blood cells varies from 20 to 80% (device performance data on file, available on request).

We believe that the lymphocytes and monocytes present in the preparation may improve healing by their effects on modulation of inflammation, tissue remodeling and repair, and phenotypic display of macrophages [19].

It was formerly shown that the difference in leukocyte concentration in PRP may affect macrophage polarization. Studies have shown that rich PRP (LR-PRP) mainly enhances the expression of M1 macrophages, while LP-PRP significantly induces the activity of M2 macrophages [20]. It was concluded that LP-PRP not only promotes cell proliferation through growth factors, but recruits repair cells through tissues and blood to promote tissue repair.

Other studies have demonstrated the beneficial effects of mononuclear cells [21,22,23]. Monocytes are associated with an increase in cellular metabolism and collagen production in fibroblasts, as well as a decreased release of antiangiogenic cytokines interferon-g and IL-12 [21,22].

Previous studies have shown that platelets activate lymphocytes to help stimulate collagen production via an increase in IL-6 expression [21,22] Thus, the current thought is that mononuclear cells may be beneficial in PRP.

Nowadays, in the clinics there is still debate whether leukocytes should be included in PRP. Some argue that leukocyte concentration should be decreased for intra-articular applications [24,25] and increased for tendon repair, for example [26].

On the other hand, granulocytes, the main population of leukocytes are pro-inflammatory cells. They are full of granules that contain potent destructive enzymes, such as peroxidases, proteases, and lysosomal enzymes. The release of these molecules is crucial for fighting bacterial infection in open wounds; however, it has a deleterious effect in aseptic lesions. It has been demonstrated in neutrophil depleted mice that the absence of neutrophils, the most abundant type of granulocytes, does not affect cutaneous healing and might even accelerate it [27]. Thus, the specific depletion of granulocytes in PRP might help to avoid undesired inflammatory reactions.

Since the full volume of plasma is recovered over the separator gel, the platelet concentration factor in PRP prepared with our devices is around 1.5 to 1.7 times the baseline value in whole blood. Thanks to the high quality of the recovered platelets, this low concentrated PRP has been shown to be therapeutically effective in all therapeutic domains in which it has been tested (see below). However, if deemed necessary, some platelet poor plasma can be discarded before the platelet resuspension step. The platelet concentration factor can thus be increased up to 3 to 4 times the baseline value in blood.

PRP with platelets of high quality at a concentration slightly above the physiological value is relevant for therapeutic use, since it does not affect tissue homeostasis, and the low level of cellular contamination (mainly lymphocytes and monocytes) reduces the risk of undesired inflammatory reactions.

It is important to note that too high a platelet concentration (more than 5 times over the baseline value) has been shown to produce suboptimal outcomes or cytotoxic effects in vitro [28,29] and in animal models [30]. Comparative clinical studies also favor less concentrated PRP [31,32]. The advantages of using highly concentrated PRP compared to PRP with a lower platelet concentration factor have not been demonstrated. The so-called therapeutic platelet concentration of 1 billion per mL (4 to 5 times over the baseline values) [33] has never been demonstrated by comparative clinical studies and might concern only leukocyte rich PRP, as a higher platelet concentration may be needed to compensate for the negative effects of pro-inflammatory white blood cells.

The know-how developed for PRP isolation with separator gel can be used for the preparation of other biologics. The gel that allows the recovery of the blood mononuclear cells (MNCs) in addition to the platelets can be used to process bone marrow aspirate for the preparation of bone marrow cell concentrates. In this tube, the bone marrow stem cells are recovered within the mononuclear cell fraction over the separator gel.

Recently, we have developed a new separator gel to isolate acellular plasma for the preparation of convalescent plasma from patients who have recovered from an infection. 


**Production of PRP in different forms to suit therapeutical needs:**


Anticoagulation with sodium citrate is reversible. Citrate prevents coagulation only by binding the plasmatic calcium ions of the blood sample used for PRP preparation. Calcium ions are essential cofactors of the coagulation cascade. Thus, when liquid PRP is injected into tissues it will coagulate thanks to the input of calcium ions by the interstitial fluid. Nevertheless, for some therapeutical applications, such as wound management, health care providers need to obtain PRP gels or clots. Citrated PRP coagulation can be induced by activators, such as thrombin, calcium solution or a combination of both.

There is often confusion between platelet activation and coagulation activation. There is a false belief that PRP should be activated to trigger the release of growth factors. This comes from early in vitro experiments, where addition of activators, such as calcified thrombin, was needed to extract the growth factors from the platelets [34]. Platelets are not simple vesicles full of growth factors but functional entities that release the growth factors in a controlled manner in response to local signals. Thus, in vitro, in the absence of activation, platelets do not release their growth factors. High doses of calcified thrombin are, thus, needed to induce complete degranulation of platelets and uncontrolled release of growth factors.

Endogenous platelet activation occurs when PRP is injected into the patient’s tissues [10]. Platelets are physiologically activated by the contact with extracellular matrix proteins (e.g., collagen) at the injection site. For each stage of the healing process, platelets secrete different growth factor cocktails, in response to local signals, to stimulate an organized tissue repair. Exogenous activation is needed only to get PRP in a gel form that coagulates rapidly at the site of injection, or to obtain a fibrin clot, autologous fibrin glue enriched with platelets or suturable fibrin membranes [35,36]. These kinds of products are used, for example, to treat difficult to heal wounds [37]. This prevents the diffusion of PRP and ensures a localized action.

We recommend the use of autologous serum that contains autologous thrombin at the physiological level to activate PRP, either alone or in combination with a pharmaceutical grade calcium solution. This serum is prepared from the blood of the patient with a specific device that also uses the separator gel technology; however, in a tube without anticoagulant. The use of autologous thrombin allows the physiological formation of a fibrin clot in which platelets secrete growth factors in a controlled and sequential manner throughout the process of clot replacement with new tissue.


**Therapeutical domains in which clinical studies have been published:**


Biologics prepared with Regen Lab devices are used in many therapeutical domains such as: Sports medicine, orthopedic surgery, skin care, and wound care, among others.

More than 150 clinical studies with successful results have been published, see Table 2 (full list available on request).

### 3.2. Innovative Biological Combinations with Hyaluronic Acid for a Synergistic Tissue Regeneration

Endogenous hyaluronic acid (HA), one of the main components of the extracellular matrix, is a polysaccharide that belongs to the glycosaminoglycan family and consists of a basic unit of two sugars, glucuronic acid and N-acetyl-glucosamine. HA serves to maintain a highly hydrated environment, regulates osmotic balance, acts as a shock-absorber and space-filler, and as a lubricant. It plays a role in cell migration and physiological angiogenesis.

Among its unique characteristics, its biocompatibility and biodegradability are very important for clinical use.

HA usually exists as a high molecular weight in the synovial fluid that surrounds joints, cartilage, and tissues of the eye and skin [38]. It is considered as a key player in the tissue regeneration process [39,40]. It has been proven to modulate inflammation, cellular migration, and angiogenesis, which are the main phases of wound healing [41]. HA biological properties are related to its molecular size: high molecular weight HA displays anti-inflammatory, immunosuppressive and anti-angiogenic properties, while low molecular weight HA has potent pro-inflammatory and pro-angiogenic molecules.

Exogenous HA can be processed and functionalized through physical and chemical modifications and crosslinking to generate versatile HA-based hydrogels presenting variable clinical applications [42]. HA has many qualities, such as moisturizing and anti-ageing effects, that recommend it over other substances used in skin regeneration [38,43]. The molecular weight of HA influences its penetration into the skin and its biological activity [43]. It may be injected intradermally or it may be used topically. HA is commonly used in other clinical applications, including intra-articular injections, ophthalmic surgery and tissue engineering (vascular, skin, cartilage, bone) [44].

**Cellular Matrix^®^**:

Many physicians are interested in combining PRP with HA. As PRP and HA target different pathways and have different functions, when used together they may have a synergistic effect as a therapeutic approach for healing, inflammation, or analgesic purposes. Cellular Matrix is the first and, so far, only CE certified medical device that allows the combination of HA and PRP in respect to medical devices and health regulations. Indeed, HA is an implantable medical device and PRP a biological drug. Consequently, healthcare professionals are not supposed to prepare their own PRP-HA mix by combining any HA with any PRP, as modifications of medical devices or biological drugs are not authorized.

This device was designed to allow the rapid and safe preparation of PRP in the presence of a high quality, non-cross-linked HA produced by bacterial fermentation, using a technology similar to the one used for PRP isolation, but with HA preloaded in the tube (Figure 3). HA creates a cell friendly matrix in which platelets are suspended. This biologically enriched network facilitates cell migration and proliferation to the treated site.


**Orthopedic use:**


Biologics, or more precisely in this case ortho-biologics, are used to treat a variety of orthopedic conditions, such as tendinopathies and osteoarthritis (OA). Numerous recent meta-analyses have demonstrated the interest of using PRP to alleviate the symptoms of knee osteoarthritis [45,46,47]. As schematized in Figure 4, the hypothesis that the association of PRP and HA would provide synergistic clinical benefit for the treatment of OA arises from the fact that the effects of PRP and HA are based on complementary mechanisms of action, biological and mechanical (viscosupplementation), respectively. This combination favors the transition from a downward spiral (“vicious circle”) in which inflammation and degeneration induce pain and joint stiffness to a situation where PRP can reverse the catabolic environment, leading to downregulation of inflammation and symptom relief. The PRP-HA combination prepared with this device has been successfully used for intra-articular injections for the symptomatic treatment of articular pain and the improvement of mobility [48,49,50,51,52,53,54,55,56,57,58,59].

Although PRP and HA have both been shown to improve pain and functional outcomes, and, when used in combination, these outcomes can be enhanced, there is some uncertainty regarding the impact of PRP and HA on cartilage regeneration, based on data from clinical trials. In fact, to visualize small changes to cartilage necessitates the use of MRI and this has often been beyond the scope of clinical trials [60]. Thus, there are only a few trials that have investigated changes in cartilage following PRP injection. So far, these studies suggest that PRP delays progression of the disease and may have a modest impact on cartilage regeneration [61,62,63,64,65,66]. Similar findings have also been reported for HA [61,67]. While similar results have been obtained in in vitro and in vivo studies, one study found that a combination of PRP and HA induced a strong recovery of cartilage surface that at the time of analysis was almost identical to the sham group, thereby demonstrating that a combination of PRP and HA could regenerate cartilage in vivo [68]. Recent studies are reinforcing those findings by the demonstration that combined therapy achieved better clinical outcomes [69,70].

Recently, we developed RegenMatrix^®^, a new all-in-one device allowing the preparation of PRP combined with cross-linked hyaluronic acid, in order to increase the residence time of HA for a longer lasting effect of the PRP and a higher mechanical effect within the intra-articular space, with, probably, a higher impact in delaying the evolution of the degenerative conditions linked to osteoarthritis.


**Dermatological applications:**


The Cellular Matrix PRP-HA combination is also used in the field of skin care for intra-dermal injections, where it has been shown to prolong the effects of PRP, thereby improving skin tone, hydration and elasticity [71,72,73].

There are several reasons for considering using a combination of PRP and HA for the treatment of skin disorders. HA is an important component of the skin. It contributes to tissue hydrodynamics by creating space for the movement of cells. It is believed to regulate the diffusion of micronutrients, metabolites and hormones between cells, and to stimulate fibroblast migration, proliferation and collagen production [74]. PRP can induce production of HA in fibroblasts [75] as well as having many other positive effects on fibroblasts and other skin cells with demonstrated clinical benefits for skin rejuvenation. Studies have shown that a PRP-HA combination is more potent than either PRP or HA alone in osteoarthritic models [68] and wound healing [76]. In vitro studies have shown that when PRP is incubated with HA, the release of growth factors is significantly increased after 5 days in comparison to PRP alone [77].


**Wound care applications:**


When PRP combined with HA coagulates, the structure of the resulting fibrin network is different and has a larger porosity than a fibrin clot without HA. This creates a better environment for cells, allowing easier cell migration and proliferation [78]. Following that evidence, we developed Cellular Matrix Wound, an all-in-one device specifically designed for wound care, which allows the easy, rapid (20 min), safe and reproducible preparation of an autologous PRP-HA clot (Figure 5) for treating wounds without the need for a coagulation activator.

## 4. Swiss Translational Research Projects for In Vitro Validation of Regen Lab Technology Platform for Tissue Engineering and Cell Therapy Research

### 4.1. Cell Therapy Research: From In Vivo to In Vitro and Back to In Vivo 

Cell therapy aims to introduce new, healthy cells into a patient’s body, to replace the diseased or missing ones. This type of therapy is challenging as it requires enough cells for transplantation into the patient. This is because specialized cells, such as fibroblast cells or brain cells, are difficult to obtain from the human body. Furthermore, specialized cells typically have a limited ability to multiply, making it difficult to produce enough cells required for certain cell therapies. Some of these issues can be overcome through the use of stem cells. Stem cells are unspecialized cells that have the ability to develop into other functional cell types. Importantly, some types of stem cells can be grown outside of the human body, thereby allowing the production of a large number of cells required for successful applications of cell therapy in medicine. When using autologous cells, the patient owns his or her own therapy.

### 4.2. Safety and Efficacy Demonstration in Translational Research Projects: Establishment of 100% Autologous Cell Culture Models

In recent years, we initiated a translational research project that highlighted the proliferative effect of PRP obtained from the patient’s own blood on adipose tissue-derived mesenchymal stem cells (ADSCs) and fibroblasts from the same patient [79,80,81] (Figure 6). Our focus is to be fully autologous in order to propose personalized therapies. 

Therefore, a line of medical devices, CuteCell, dedicated to in vitro human cell culture in research laboratory settings (GMP facilities) was developed. These devices allow the preparation of PRP (CuteCell-PRP), serum (CuteCell-serum) and MNC enriched PRP (CuteCell-MNC). These new products offer an autologous alternative to fetal bovine serum (FBS) in culture media for cell therapies [82].

#### 4.2.1. Adipose-Derived Mesenchymal Stem Cells (ADSCs)

-Enhancement of ADSC expansion with PRP supplementation

To define an autologous system for ADSC proliferation, we assessed the efficiency of autologous PRP on ADSC proliferation in comparison to the classical FBS-supplemented medium. We investigated the optimal PRP concentration in the culture media. Culture media supplemented with PRP prepared from the patient’s own blood enhanced by 14 times the in vitro proliferation of ADSC, in comparison to the classical culture medium prepared with 10% FBS over 10 days [79] (Figure 7).

-PRP does not alter ADSC phenotype, differentiation capacity, and chromosome status

We verified that the intrinsic differentiation potential of ADSC into adipocytes, osteocytes, or chondrocytes was kept intact regardless of the supplements used for their proliferation. We observed that PRP did not adversely affect the differentiation capacity of ADSC. Adipogenic, chondrogenic, and osteogenic differentiation potentials were confirmed and were qualitatively comparable to 10% FBS [79]. Cytogenetic analysis of cells, cultured in either 10% FBS or 20% PRP conditions, did not show any abnormal karyotypes. FBS and PRP culture conditions had numerical and structural stability. Thus, treating cells with PRP did not modify the chromosomal stability [79].

#### 4.2.2. Human Dermal Fibroblasts (NHDF)

Nowadays, autologous fibroblast application for skin repair presents an important clinical interest. In most cases, in vitro skin cell culture is mandatory. However, cell expansion, using xenogeneic or allogenic culture media, presents some disadvantages, such as the risk of infection transmission or slow cell expansion.

-Enhancement of fibroblast expansion with autologous PRP treatment

In this study, we retrieved fibroblasts and blood from the same patients. After 7 days of culture, fibroblasts supplemented with different PRP concentrations showed a higher viable cell number compared with FBS-containing media (Figure 8). This proliferative effect of PRP followed a dose-dependent bell-shaped curve. The optimal culture condition was PRP 20%, where the NHDFs number was 7.7-fold higher than FBS 10% [80,81]. 

-Optimal PRP concentration is crucial for the maintenance of fibroblast phenotype

Fibroblasts cultured in a classical FBS-supplemented culture medium showed a regular flattened cell shape while those treated with PRP (10–50%) were spindle shaped, a morphology that is closer to 3D matrix cultures or in vivo setting. We sought to investigate whether the morphological change occurring at 7 days of PRP treatment was related to a phenotypical change. We first demonstrated prominent F-actin reorganization from cortical actin localization (FBS 10%) into thick cell-spanning filaments (PRP 20%). We further assessed the changes in alpha-SMA expression upon PRP treatment by flow cytometry and immunofluorescent analyses. Alpha-SMA expression was significantly increased with a high PRP concentration (40–50%), while FBS-10% and PRP 5–10%-treated cells showed a basal perinuclear staining. Immunofluorescent analysis showed an increase in vimentin staining in the presence of PRP 20%, but it was completely abolished at high PRP concentration (PRP 50%) [81]. 

These results underscored the importance of optimal PRP concentration. They confirmed that high platelet concentration is no better than moderate concentration and could even be harmful [28,29].

-CuteCell PRP modulates metabolic activity, fibroblast adhesion, and favors migration

Using an MTT metabolic test, we evidenced an increase in PRP-treated cells, directly reflecting an increase in cell metabolic activity peaking at 3.12-fold in PRP 20%-treated cells compared with FBS 10%-treated cells after 48 h of treatment. To further characterize the biological effects of PRP on fibroblast biology, we evaluated the effect of PRP treatment on cell adhesion on laminin and collagen type I. PRP decreased the overall attachment of fibroblasts to laminin 4 h after seeding. This effect had already occurred after 15 min, with a 21% decrease in overall cell adhesion. The same results were obtained for fibroblast attachment to collagen I matrix (41% of total cell adhesion after 15 min). To study the migratory properties of fibroblasts exposed to PRP, we performed an in vitro scratch assay (Figure 8). Eight hours of 20% PRP treatment induced a 10% increase in the number of migrating cells from the scratch margin into the scratch zone compared with cell cultures with FBS. This migration front was a collective cell migration. Conversely, fibroblasts exposed to FBS 10% showed features of isolated cell migration [81].

-Whole genome analysis to demonstrate that CuteCell PRP is safe at the genomic level

To document genetic stability during proliferation, we cultured NHDFs for 4 days with media supplemented with FBS 10% or PRP 10%. Array CGH analysis of cells treated with the two different culture media did not show imbalanced chromosomal rearrangements. The increased proliferation rate in response to PRP treatment did not provoke genomic instability [81]. These results are of prime importance as some studies claim that growth factors released by PRP can contribute to tumor progression [83].

### 4.3. Angiogenesis Is Differentially Modulated by Platelet-Derived Preparations 

In tissue regeneration, therapeutic angiogenesis aims at restoring a proper vascular system through the delivery of exogenous growth factors, cytokines and chemokines, among others. Important angiogenic factors are VEGF, angiopoietin, FGF, HGF, PDGF, and TGF, although a myriad of other proteins is also known to be involved in blood vessel formation. The release of platelet-derived products, such as autologous growth factors, cytokines and chemokines, can trigger therapeutic angiogenesis. In an in vitro study, we evaluated and compared the ability of three platelet-derived preparations: platelet-rich-plasma (PRP), PRP hyaluronic acid (PRP-HA) and platelet lysates (PL) at various concentrations (5–40%) to modulate human umbilical vein endothelial cells’ (HUVECs’) biological effects on metabolism, viability, senescence, angiogenic factors secretion and angiogenic capacities in 2D (endothelial tube formation assay or EFTA) and in 3D (fibrin bead assay or FBA) [84].

-PRP and PRP-HA modulate endothelial cells (HUVEC) angiogenic activities in 3D

In this work, we used a high throughput in vitro 3D angiogenesis assay (the fibrin bead assay or FBA) to model the angiogenic effect of platelet-derived preparations used in tissue engineering studies or in the clinic (Figure 9A). We tested a range of concentrations (5 to 40%) of standardized CuteCell PRP, CM-PRP-HA, obtained with a Cellular Matrix BCT-HA tube, and a commercial preparation of platelet lysates. Blood samples were obtained from healthy donors. The procedure conformed to the principles of the Declaration of Helsinki and was approved by cantonal Geneva ethics and research commission (ID 2017-00700, approved on 18 October 2018). The FBA uses a culture of endothelial cells (ECs) on the surface of ~200 μm-sized Cytodex-3 microspheres embedded in a 3D bovine fibrin matrix, with normal human dermal fibroblasts (NHDFs) used as feeder cells. These stromal cells provide various angiogenic growth factors: hepatocyte growth factor (HGF), transforming growth factor alpha (TGF-α), angiopoietin-1 (Ang-1), as well as matrix molecules, matrix-modifying proteins and matricellular proteins (e.g., procollagen C endopeptidase enhancer 1, secreted protein acidic and cysteine-rich (SPARC), transforming growth factor-β-induced protein Ig-H3 (βIgH3) and insulin-like growth factor binding protein 7 (IGFBP7)) [85]. In control conditions, sprouting of neo-vessels was apparent between days 2 and 3, and cultures were imaged on day 4. For the quantification of micro-vessel network sprouting, samples were automatically scanned with a high-throughput imager and analysis of the microsphere images was performed using a method we developed with ImageJ software (Figure 9B) [86,87]. This assay represents a significant improvement over conventional, single-cell-type angiogenic assays, as the inclusion of multiple cell types more closely mimics the physiological environment [85]. 

The basic steps of sprouting angiogenesis include enzymatic degradation of the capillary basement membrane, endothelial cell (EC) proliferation, directed migration of ECs, tubulo-genesis (EC tube formation), vessel fusion, vessel pruning, and pericyte stabilization. Platelet-derived preparations elicit different angiogenic responses when tested in the FBA where they act on different steps of the angiogenic process. Thanks to the powerful automatic quantification method we have developed [86], we can assess the modulation of the morphometrical parameters of the neo-vessels formed in the fibrin matrix. Examples are the total length of the microvascular network, the anastomosis in the vessels (branches), the number of capillaries arising from the beads (anchorage), and the number of vessel tips showing the complexity of the network (extremities) (Figure 9B).

In our study, the most potent preparations were PRP and CM-PRP-HA, as the total length of the neovascular network, total length of the branches, extremities and anchorage were highly stimulated, compared to the control conditions (VEGF± heparin treatment) (Figure 10). Platelet preparations stimulated all steps of the angiogenic process, as massive sprouting of a branched micro-vessel network was observed by optical microscopy (Figure 10A). PRP was the most potent angiogenic preparation, significatively stimulating angiogenesis from 2 to 12-fold, depending on the concentration of PRP used and the parameter of interest (Figure 10C). Significant angiogenesis was already observed with the lowest PRP concentration (5%). CM-PRP-HA also stimulated angiogenesis but to a lesser extent than PRP: this could be explained as being due to the concentration of platelets in this preparation being reduced by half in comparison to PRP. Platelet lysates had to be highly concentrated to elicit the same angiogenic response as PRP and CM-PRP-HA [84].

-Effect on HUVEC viability and senescence

To assess cell viability, we added Crystal violet as a dye to study the effect of platelet-derived products on cell viability (Figure 1C). After 3 days of treatment, more viable violet cells were found in PRP and PRP-HA-treated cultures. PRP and PRP-HA (10–40%) increased cell viability in a dose-dependent manner compared to the control condition. PL also increased cell viability, but to a lesser extent [84].

To evaluate the effect of platelet-derived preparations on HUVEC senescence, we induced aging in vitro by cultivating the cells at a low density [88] from passages 2 to 6. In vitro aging induces an increase in size for cells and their nuclei, and more apoptotic cells appear [89]. In this assay, we tested 10% FBS, Heparin 2 UI/mL, 10% PRP, 10% PRP-HA and 10% PL. Long-term serial cultivation in FBS or heparin media induced senescent cells, while no senescent cells were observed with platelet-derived treatments [84]. 

## 5. Conclusions

Regenerative medicine encompasses a wide range of techniques aimed at repairing, or even replacing, damaged or aged tissues. Among them, autologous platelet-rich plasma is one of the simplest and most efficient ones. This approach is based on the intrinsic abilities of the human body to repair itself and the role of the platelets in this process. There is a growing interest in using standardized PRP, alone or in combination, in regenerative medicine because it represents a safe and natural treatment, and it has, so far, demonstrated promising results in a large number of therapeutic indications. 

Many medical devices for PRP preparations are on the market. They vary greatly in terms of technology and PRP final composition. We discussed their specificities and limitations in comparison to our technology.

The CuteCell medical device was designed and validated for in vitro cell culture in autologous conditions that respect GMP guidelines and the standards of regulatory agencies. CuteCell-PRP was demonstrated to be an efficient, cost-effective, and safe biologic supplement for adipose-derived stem cells of fibroblast cultures, as well as a substitute for xenogeneic or allogenic blood derivatives for the validation of future clinical protocols of in vitro cell expansion. We showed that platelet-derived products are autologous biologics that drive angiogenesis in situ without the need for pre-vascularized exogenous materiel engraftment. PRP and PRP-HA were the only preparations allowing the whole angiogenic process to complete.

PRP is now a key player in the medical world with millions of patients treated every year. However, limitations related to the lack of a standardized procedure for its preparation make it difficult to compare available clinical data. The technology described in this publication provides a solution to the challenge of standardizing PRP preparations intended for therapeutic uses.

## Figures and Tables

**Figure 1 pharmaceutics-14-01617-f001:**
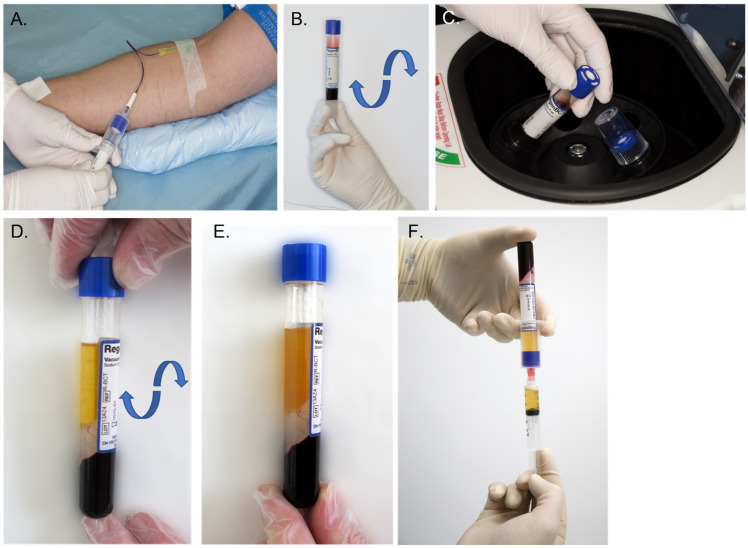
Process of standardized PRP production. Regen PRP tubes are designed for a use in a closed circuit to maintain the sterility of the biological samples. (**A**) The venous puncture is performed, and the desired number of Regen PRP tubes are filled with whole blood. The vacuum within the tubes enables automatic collection of the necessary volume of blood (about 10 mL). (**B**) The tubes are carefully reversed upside down three times to mix the blood with the anticoagulant. (**C**) The tube is centrifuged with a relative centrifugal force of 1500 g for 5 to 9 min. (**D**) After centrifugation, the blood is fractionated; the red and white blood cells are trapped under the gel, and platelets settle on the surface of the gel. (**E**) The tube is gently rocked to re-suspend the platelets. (**F**) The resulting PRP is collected with a syringe connected to a transfer device.

**Figure 2 pharmaceutics-14-01617-f002:**
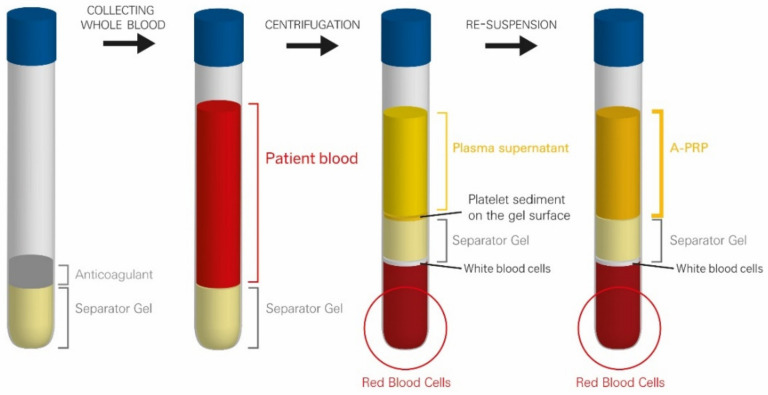
Principle of operation of Regen Lab technology for PRP preparation.

**Figure 3 pharmaceutics-14-01617-f003:**
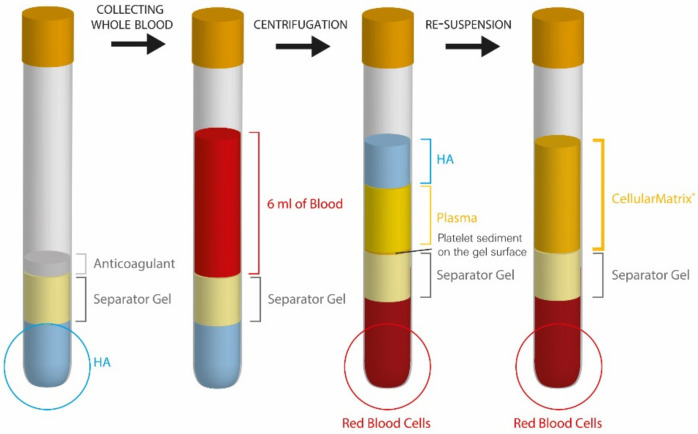
Different steps of preparation of the PRP-HA mix with Cellular Matrix tubes.

**Figure 4 pharmaceutics-14-01617-f004:**
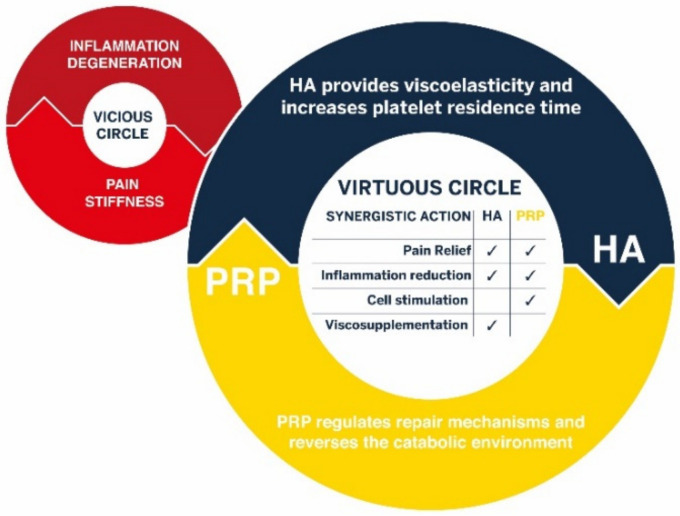
Complementary mechanisms of action of HA and PRP provide synergistic benefit for the treatment of osteoarthritis and other degenerative joint diseases.

**Figure 5 pharmaceutics-14-01617-f005:**
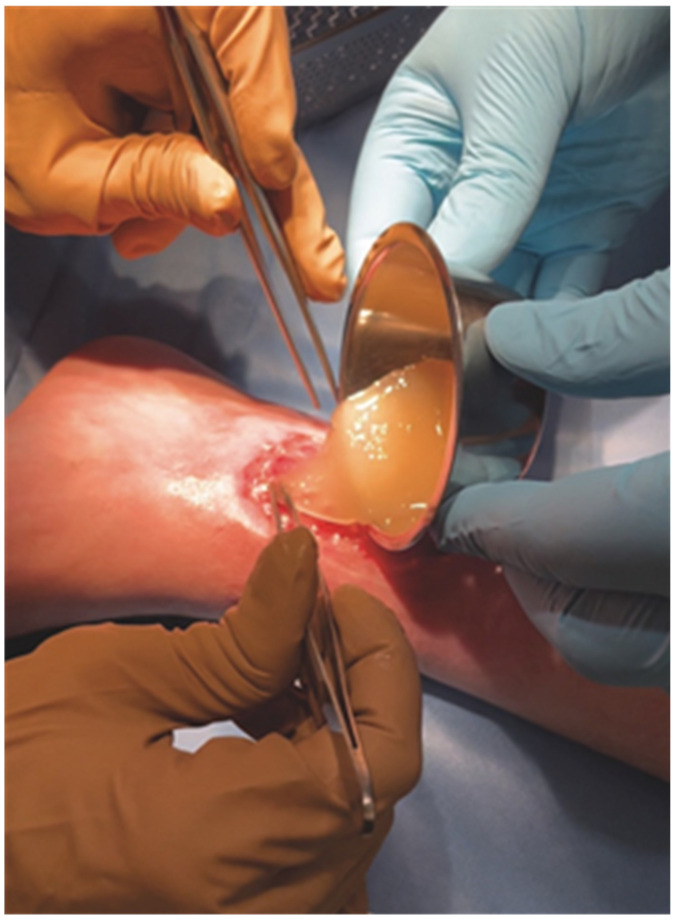
PRP-HA gel applied on a disinfected wound as a biological dressing.

**Figure 6 pharmaceutics-14-01617-f006:**
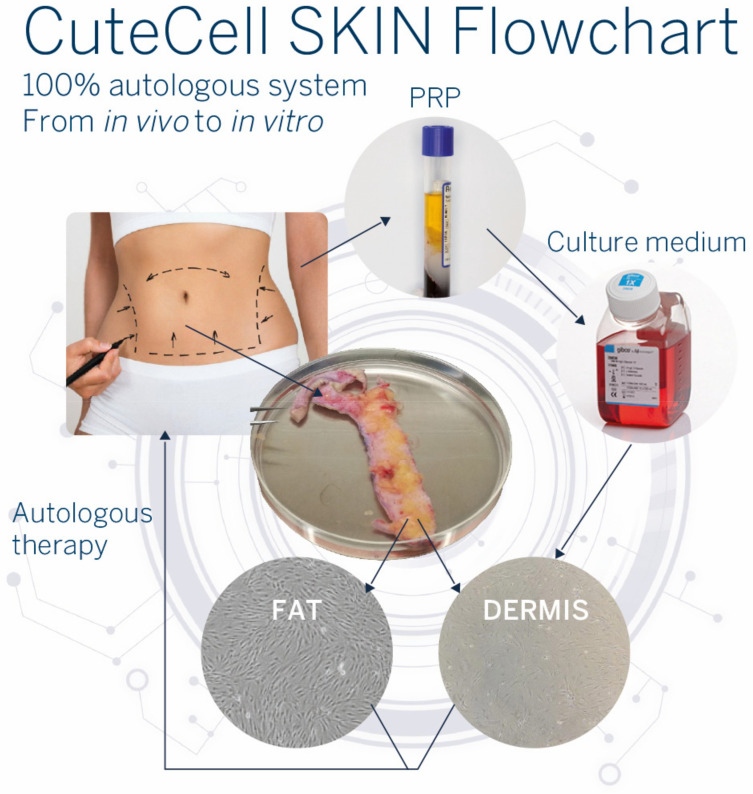
CuteCell SKIN flowchart. During surgical abdominoplasty, a piece of skin is retrieved from the patient as well as blood from venous puncture. In the laboratory, cells from skin (fat or dermis parts) are isolated for in vitro cultures. The patient’s blood is processed with the CuteCell PRP device. The prepared PRP can be added to the culture medium in replacement of FBS to expand the patient’s cells in vitro.

**Figure 7 pharmaceutics-14-01617-f007:**
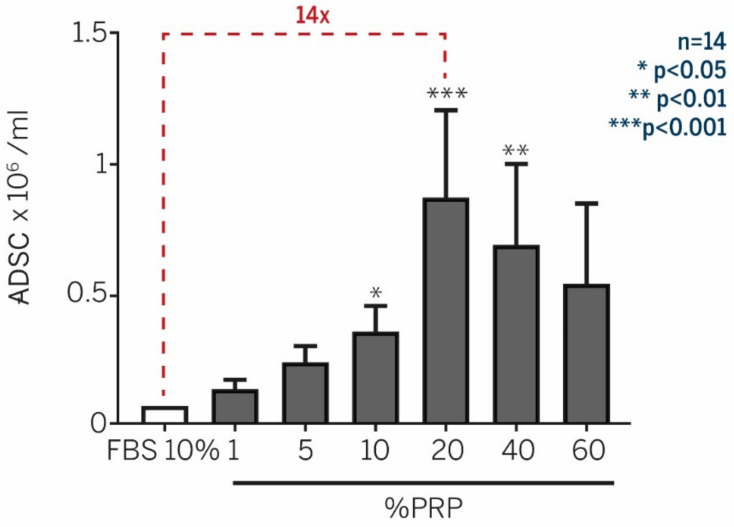
Culture media supplemented with autologous PRP drastically enhanced in vitro proliferation, in comparison to the classical culture medium with 10% FBS. Data from Atashi et al. Tissue Engineering. Part C. 2014 [79].

**Figure 8 pharmaceutics-14-01617-f008:**
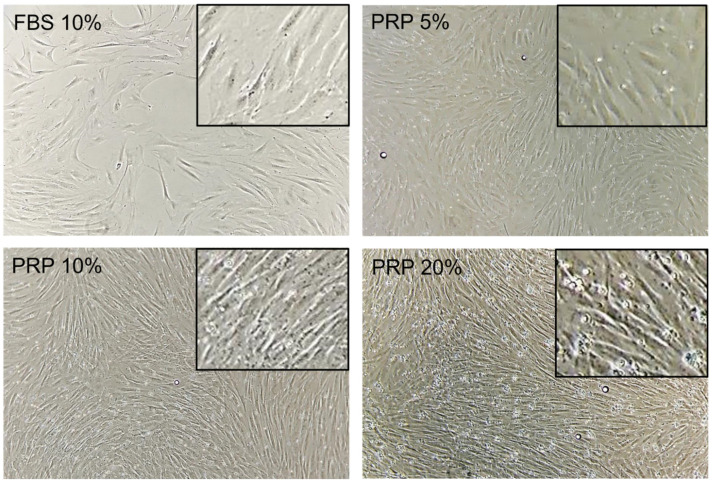
Bright-field optical photography of NHDF in the presence of FBS 10% or PRP (5–20%) after 7 days of culture. Magnification 10×. Pictures are representative of one donor. Data from Berndt et al. Tissue Engineering. Part A. 2019 [81].

**Figure 9 pharmaceutics-14-01617-f009:**
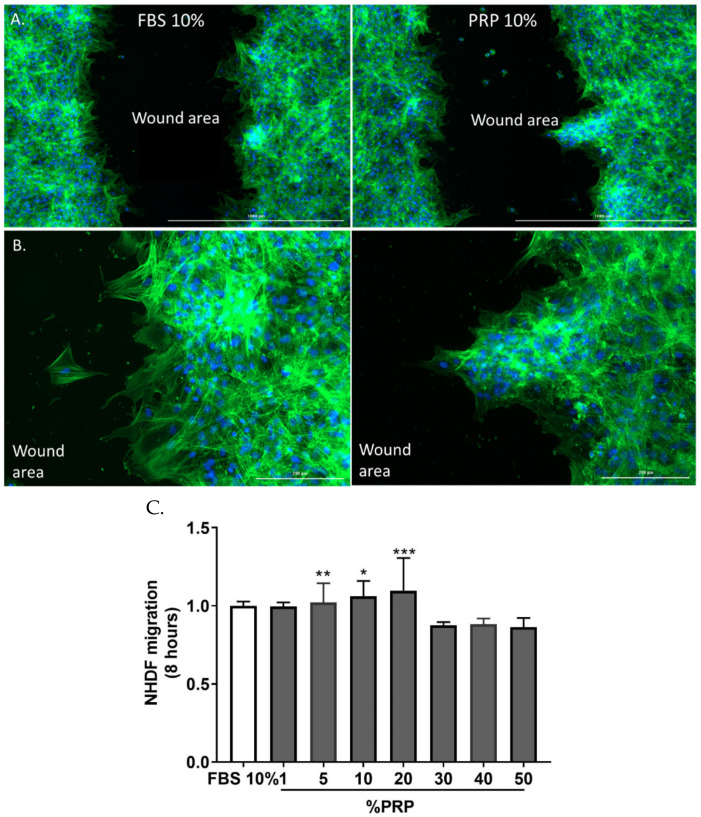
Comparative cellular effects of PRP 10% treatment on cell migration in fibroblast cultures. (**A**) Migrating fibroblasts narrowed the width of the scratch zone as evidenced after 8 h with immunofluorescence staining with phalloidin. The cell migration front was equally distributed along the scratch border in FBS 10%, while it was less homogeneous in PRP 10%-treated cells. (**B**) Zoom in pictures showing isolated cell migration in FBS 10% cultures and collective cell migration in PRP 10% cultures. Data from Berndt et al. Tissue Engineering. Part A. 2019 [81]. (**C**) Quantification of migration using Image J software (* *p* < 0.05, ** *p* < 0.01, *** *p* < 0.005).

**Figure 10 pharmaceutics-14-01617-f010:**
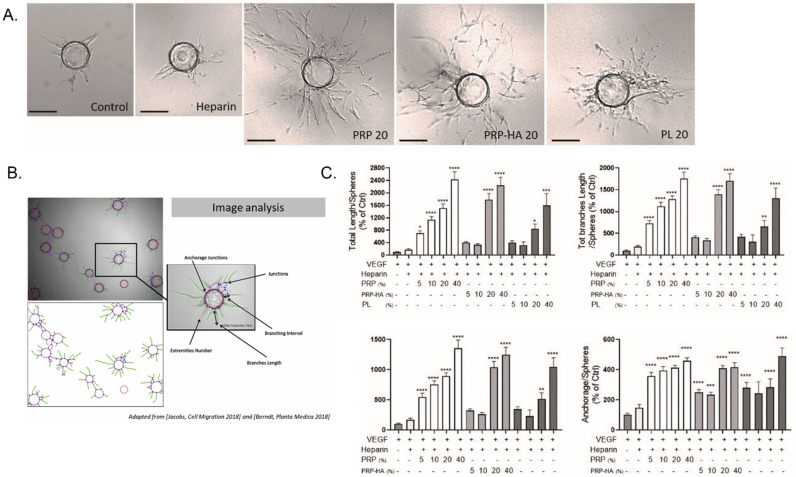
Platelet-derived products differentially modulated angiogenesis in the 3D fibrin bead assay. (**A**) 3D HUVEC cultures were treated with platelet-derived products (PRP, PRP-HA and PL) at different concentrations for 4 days. Representative pictures of massive enhancement of angiogenesis (PRP 20, PRP-HA 20) or slight endothelial proliferation from the EC coated beads (PL20) compared to control conditions (control and heparin) at day 4. Scale bar: 150 μm. (**B**) Representative images of enhanced angiogenesis. Cytodex microcarrier beads coated with HUVECs exhibiting pseudo-capillary growth after automatic analysis with a specific plugin developed for the Image J software [87]; some of the morphometrical parameters of interest are shown. (**C**) Quantification of morphometrical parameters of the capillary network was performed by a computerized method (thanks to Image J opensource software) on pictures taken on day 4. Representative parameters measured were total length, total length of branches, number of extremities and number of anchorage junctions per sphere. Graphs are representative of three independent experiments. One hundred spheres were quantified for each experimental condition and compared to the heparin-treated cultures. Significant from heparin as measured by the one-way ANOVA followed by Dunett’s multiple comparison test (* *p* < 0.05, ** *p* < 0.01, *** *p* < 0.005, **** *p* < 0.001). Data from Berndt et al. Biomedicines 2021 [84].

**Table 1 pharmaceutics-14-01617-t001:** Examples of medical devices for PRP preparation. Blood Vol.: Volume of blood PRP vol.: Volume of PRP; Platelet conc. factor: Platelet concentration factor.

Technology	Examples of Devices	Blood Vol.	PRP Vol.	Platelet Conc. Factor	Biological Product	Remarks
Low force centrifugation in tubes or syringes	BTI PRGF Endoret	8 mL	1–2 mL	2–3×	**Plasma PRP** **(LP-PRP)**	Devices with no physical separation of platelet rich plasma from red and white blood cells. **Operator dependent results**
	Arthrex ACP double syringe	15 mL	4–6 mL	1.6–2×		
High force centrifugation in hourglass shaped devices	Tozai Holdings Prosys	25 mL	2 mL	6–8×	**Buffy coat PRP** **(LR-PRP)**	
High force centrifugation in devices with floating buoy or shelf	Biomet GPS III	27–54 mL	3–6 mL	6–9×	**Buffy coat PRP** **(LR-PRP)**	Devices with physical separation of platelet rich plasma from red and white blood cells. **Device dependent results**
	Harvest SmartPrep2	27–54 mL	3–7 mL	3–8×		
Variable force centrifugation in computer aided systems	Arthrex Angel	40–180 mL	variable	1–18×	**Setting dependent product**	
	Arteriocyte Magellan	26–52 mL	6 mL	3–7×		
High force centrifugation in tubes with separating gel	RegenTubes	10 mL	5–6 mL Can be reduced to 1.5 mL	1.6–1.7× Up to 4× by PPP removal	**Plasma PRP** **(LP-PRP)**	

**Table 2 pharmaceutics-14-01617-t002:** Published clinical studies in which biologics manufactured with Regen Lab devices were successfully used.

Herapeutical Domain	Pathologies	Studies	Treated Patients
Sports medicine	Osteoarthritis and other cartilage pathologies	37	2053
	Tendinopathies	15	596
	Muscle	3	18
	Carpal tunnel	6	215
Orthopedic surgery	Bone reconstruction, cartilage and tendon surgery	16	649
	Spine surgery	3	46
Dermatology	Skin care	14	353
	Alopecia	10	240
Chronic wounds		12	238
Plastic and reconstructive surgery		7	178
Cardiovascular surgery wounds		3	425
Urology		4	122
Gynecology		8	105
Penile disfunction		3	151
Fertility		3	67
Ophthalmology		10	170
Otorhinolaryngology		7	103
Dental/maxillofacial surgery		7	200
Total		168	5929

## Data Availability

Not applicable.

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
