# Peer review of "Swiss Medical Devices for Autologous Regenerative Medicine: From Innovation to Clinical Validation"

_pharmaceutics, 2022, doi:10.3390/pharmaceutics14081617_

Round 1
Reviewer 1 Report
The preparation of autologous blood-derived products is an interesting topic because of multiple applications, and because there is still lack of the standardization of PRP preparation methods. However, this work has some flaws and weakness. First, this is not a review, because lack references to main points and obtained results were not carefully compared to other works. The main objection is that this review is not based on the existing literature (references to the state of the art). Thus, if it is not meant to be just the only advertising text, it requires major revision.
Other comments;
Please indicate the aim of the study.
Line 56: please add the reference
Line 175 Reference? Or Authors own statement?
Line 181; were previous experiments conducted?
Line 193; references or Authors previous experiment, please add this information?
Lines 194-199; previous experiments were conducted?
Please shortly describe the equivocal role of WBC as a component of PRP and differences is applications of L-PRP and PRP without leukocytes on the basis of current literature.
Before chapter 3.2 please shortly describe the properties of HA, the role of this compound in tissue repair and previous findings on this topic (for example from references 42, 43, 44, 45, 47, 48).
Figure 5; please add the arrows indicating “flow” and please add the arrow back to the patient as the final of this procedure (clinical application?)
Please change the structure of the article, according to the title “from innovation to clinical validation”. In the present form it is not clear enough.
The conclusion does not reflect the content.
Reviewer 2 Report
The authors Gomri et al. who are all affiliated with the firm Regen Lab SA describe the scientific thought on the development and quality control of their products that deliver platelet-rich plasma. Moreover, they reviewed and summarized literature that depicted data using their technique to gain platelet-rich plasma. They show results using different in vitro assays to e.g. assess the influence of platelet-rich-plasma on enhancing amounts of adipose-derived stem cells, gaining regenerative function of fibroblasts and endothelial cells. They conclude that their technology "provides a solution to the challenge of standardizing PRP preparations intended for therapeutic uses".
Overall, to be scientifically sound, other technical approaches in regenerative medicine and even regarding platelet-rich plasma must be introduced and discussed more thorougly. In detail, the close affiliation and sole description of in vitro studies depicting research conduced with Regen Lab platelet-rich plasma bears high risk that the conclusion is misleading. Strong exmaples are the findings in Figures 6-8 because 10% FBS solution is not an ideal (positive) control to compare to the influence of platelet-rich plasma in vitro whereas a growth medium formulation should have been used to compare these findings.
The approach explaining the development and quality control of the products is interesting and English is fine.
Reviewer 3 Report
This is an interesting manuscript and discusses crucial points about PRP.
But the abstract needs to be reformulated: include the points that will be discussed in the manuscript.
Line 66: The authors cited two types of PRP. It needs to be cited.
Line 144-153. Please include a scheme of this kind of tubes and sterility approach.
Line 203-204: Cite the types of separator tubes, and include references.
Line 316-319: Include newer references.
Line 366-371: Include references.
Figure 7: Increase the image and include a figure showing the live cells.
Figure 8: Include into the figure the respective graph showing the migration of the cells.
Figure 9: It is not necessary for this manuscript. Part of it can be included into the Figure 10.
The abbreviations list needs to be update.
Paragraph spacing need to be reviewed.
Include the ethical committee for this study.
Round 2
Reviewer 1 Report
Authors made major revision and in my opinion the manuscript deserves publication in present form.
One comment, line 302 collagenases belong to proteases, please change this part.